


# Water vapour fluxes at a Mediterranean coastal site during the summer of 2021: observations, comparison with atmospheric reanalysis, and implications for extreme events.

Fabio Madonna[1,2], Benedetto De Rosa[2], Simone Gagliardi[2], Ilaria Gandolfi[2], Yassmin Hesham
Essa[3,4], Domenico Madonna[5], Fabrizio Marra[2], Maria Assunta Menniti[6], Donato Summa[2], Emanuele
Tramutola[2], Faezeh Karimian Saracks[1], Filomena Romano[2], and Marco Rosoldi[2]

[1]Department of Physics, University of Salerno, 84084, Fisciano, Italy.
[2]Consiglio Nazionale delle Ricerche, Istituto di Metodologie per l'Analisi Ambientale (CNR-IMAA), 85050, Tito Scalo,
Italy.
[3]Goethe University Frankfurt, Institute for Atmospheric and Environmental Sciences (GUF-IAU), 60438, Frankfurt am
Main, Germany
[4]Agricultural Research Center, Central Laboratory for Agricultural Climate (ARC-CLAC), Giza, Egypt
[5]Istituto Salesiano Sant'Antonio di Padova, Soverato, Italy.
[6]Centro Studi e Ricerca Ambiente Marino (CESRAM), Soverato, Italy.

*Correspondence to*: Fabio Madonna (fmadonna@unisa.it)

**Abstract.** Extreme events in the Mediterranean have increased in frequency and intensity during the last decade, and this
trend is expected to continue in the future. Characterising the features of extremes using observations and well-established
datasets is critical for understanding the processes and development of extreme weather, as well as improving forecast.
This study uses observational data collected during the Mediterranean Experiment for Sea Salt And Dust Ice Nuclei
(MESSA-DIN) from July to September 2021 to characterise the synoptic analysis of the severe summer of 2021. The
analysis focuses primarily on water vapour fluxes, humidity, convective parameters, and the role of aerosols in cloud
formation. Furthermore, we compare the findings to the widely known atmospheric reanalysis ERA5, pointing out
agreements and inconsistencies with observations, as well as discussing aspects that can improve modelling activities for
addressing and forecasting extreme weather, with a focus on the extreme flooding that affected Central Europe in July
2021. The findings show the crucial role of water vapour fluxes in regional climate events, emphasizing the need for high-
resolution data from microwave radiometers and atmospheric profilers to verify and improve predictions under complex
atmospheric conditions. ERA5 performed rather well in synoptic representation, but it exhibited a dry bias in RH values,
which affected the accurate representation of water vapour fluxes. By comparing the ERA5 RH bias with upper-air
measurements available during the campaign period at the Potenza GRUAN (GCOS Reference Upper-Air Network) site,
in South Italy, the bias was further examined, showing to exhibit an irregular behavior at different sites. The findings also
emphasize the need for improving reanalysis model performance in complex terrain conditions, particularly near coastal
areas, as well as the use of km-scale models for mesoscale research and dealing with extreme weather.

## 1 Introduction

The Mediterranean summer is often characterized by intense water vapour fluxes due to intense evaporation, feeding the
zonal or meridional air mass transport and representing a significant source during extreme precipitation events (Ciric et
al., 2018). Severe precipitation in the Mediterranean Basin, although generated only by the contribution of remote water
vapour sources, required moisture uptakes associated with anomalously intense evaporation (Winschall et al., 2014).





Water vapour fluxes, primarily originating from the Atlantic, North Africa, and regional seas, contribute to the high
relative humidity observed in the troposphere, especially during the summer months. These fluxes have profound effects
on regional climate boosting intense rainfall events in certain regions while prolonged droughts in others. Persistent water
vapour fluxes also influence the region's radiative balance, exacerbating the effects of surface radiation trapping and
amplifying heatwaves' effects. Therefore, understanding the role of water vapour fluxes in Mediterranean is key for
predicting extreme weather events, such as the severe floods and heatwaves that periodically affect the Mediterranean
(Russo et al., 2017).

During the summer of 2021, which was one of the warmest on record for Europe among the last decades (Lhotka and
Kyselý, 2022), several regions, particularly in the Mediterranean basin, experienced severe soil moisture deficits.
Southwestern Europe faced heatwaves in June, July-August, and September, with monthly average temperatures slightly
below the warmest summer in Europe of 2022 as reported in the European State of the Climate (ESOTC) and recent
studies, but with the warmest temperatures on record in South Italy (ESOTC, 2023; Gandolfi et al., 2024; Merlone et al.,
2024). The dry conditions observed in the northern Mediterranean basin extended to northern Tunisia, while average soil
moisture levels in other areas of Northern Africa along the coastline were from average to higher than average compared
to climatology values (ESOTC, 2021).

Using a combination of ground-based measurements, from the mobile facility of the Atmospheric Observatory of the
Institute of Methodologies for Environmental Analysis of the Italian National Research Council (CNR-IMAA), CIAO
(Madonna et al., 2010), and the fifth-generation of reanalysis data, ERA5 (Hersbach et al., 2020; Essa et al., 2022), this
study investigates the effect at a coastal site of the frequent enhancement of water vapour fluxes due to intense evaporation
of the Mediterranean Sea in the summer of 2021. The measurements were collected in the frame of the Mediterranean
Experiment for Sea Salt And Dust Ice Nuclei (MESSA-DIN), held in Soverato, South Italy (Latitude: 38.6894°N,
Longitude: 16.545278°E, 30 m a.s.l.). One of the main goals of the campaign was the study of aerosol-water vapour-
cloud interactions, with a focus on sea salt and dust. A ground-based remote sensing facility was operated at this coastal
site from June 24th to November 8th, 2021. In this paper, we discuss the results of the water vapour and measurements of
equivalent blackbody sky brightness temperature, collected with a microwave radiometer and an infrared thermometer,
24h/7days during the campaign in the summertime, from June 24th to September 30th. The time series showed frequent
high relative humidity values in the mid-troposphere, investigated to identify the contribution by the contribution of water
vapour fluxes and convection, also through a synergic data analysis involving the radiometer, a sun photometer, aerosol
lidars and a cloud radar, as well as the comparison with ERA5 reanalysis. The analysis was also extended to verify the
role of aerosols in the local paucity of warm and cold cloud layers during the period of the campaign. Additionally, the
paper discusses the potential correlation of water vapour fluxes enhancement over the Mediterranean Sea with severe
weather events, such as the flooding that occurred in July 2021 in Central and Eastern Europe, illustrating the critical
importance of increasing the amount of ground-based water vapour measurements in the Mediterranean in support of
accurate water vapour flux predictions for effective forecasting of extreme rainfall and flood events.

The paper offers in section 2 an overview of the instruments and datasets used during MESSA-DIN. Section 3 presents
the analysis of the ground-based measurements and the related results, including the comparisons with reanalysis, the
synoptic study of water vapour transport using ERA5 data, and the role of aerosol in the local cloud formation. In the last
section, discussion of results and conclusions are provided.



## 2  Instruments and datasets

Figure 1 shows an aerial map of the Soverato with an indication of the measurement site where the MESSA-DIN campaign
80  took place.

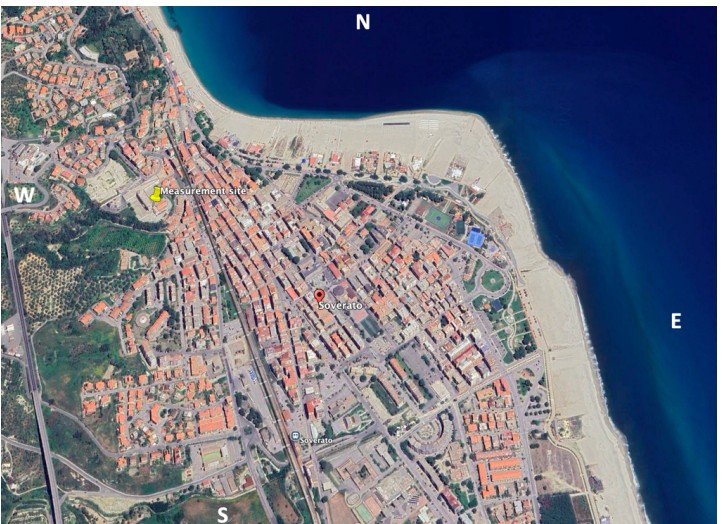

Figure 1: Aerial photo of the measurement site in the town of Soverato where the MESSA-DIN campaign was carried
out. Cardinal directions are also indicated. Image obtained from Google Earth (version 7.3.3.7786), Accessed on
September 28, 2024, Coordinates: 38.6894°N, 16.545278°E. © Google, images © 2024 Maxar Technologies.

85  For studying aerosol, water vapour and clouds, the measurement site was equipped with several instruments, including a
Ka-band Doppler radar, laser ceilometer, a UV polarization Raman lidar, a microwave radiometer complemented by an
infrared radiometer, a sun photometer, a wind doppler lidar, a total sky imager, and other near-surface measurements. In
this section, the instrument used in the data analysis presented in this paper are shortly described.

The CL51 ceilometer provides cloud base height and an attenuated backscattering coefficient profile up to 15 km above
90  ground level (a.g.l.), with maximal vertical and time resolution of 10 m and 10 s. It uses a pulsed diode laser source
emitting at $910 \pm 10$ nm with a repetition rate of 6.5 kHz and a refractor telescope to collect the backscattered radiation
from the atmosphere. Limitations include a low signal-to-noise ratio (Madonna et al., 2018). CL51 measurements
discussed in this paper have been processed using the ACTRIS-Cloudnet retrieval algorithm (Rosoldi, 2024).

The 36 GHz Ka-Band (MIRA-36) is a monostatic, magnetron-based pulsed Ka-Band Doppler polarimetric radar for
95  unattended long-term observations of cloud properties. It provides range-resolved measurements of reflectivity, vertical
wind, and linear depolarization ratio (LDR) from clouds (Nickovic et al., 2016) and giant aerosols (Madonna et al., 2010;
Madonna et al., 2013), up to 15km, with typical range and time resolution of 30 m and 30 s,

The Doppler lidar (Halo Photonics Stream Line XR) emits laser pulses at 1.5 μm with repetition frequency of 10 kHz. It
is capable of full upper hemisphere scanning with 0.01° resolution in both azimuth and elevation and is equipped with a
100  heterodyne detector, measuring range-resolved elastic backscattered radiation from aerosol and cloud particles, as well



as their Doppler radial velocity (along the line of sight) up to 12 km for clouds, and depending on the aerosol load in a cloud-free atmosphere. The Doppler velocity precision is less than 20 cm s⁻¹ for SNR greater than -17 dB. During the campaign, the lidar was mainly operated in zenith pointing configuration with its maximal range and time resolution of 30 m and 1 s, respectively. Moreover, every 5 minutes the system performed a conical scan with 6 off-zenith beam directions, elevation angle 75°, azimuth angles equally spaced of 60° and time averaging for each direction of 3 s. The vertical profiles of horizontal wind speed and direction are retrieved with vertical resolution of about 30 m and time resolution of 5 minutes from Doppler velocity profiles measured during conical scans, using the algorithm described in Päschke et al. (2015).

The Radiometrics Microwave Profiler (MWP) 3014 measured integrated water vapour (IWV), liquid water path (LWP), zenith brightness temperatures, and thermodynamic profiles (Cimini et al., 2018; Madonna et al., 2010) from the beginning of the campaign until September 30ᵗʰ, when it stopped due a failure of the power supply unit. Integrated Water Vapour (IWV) and Liquid Water Path (LWP) are derived from brightness temperatures at 22.235, 23.035, 23.835, 26.235, and 30.000 GHz. Temperature profiles were derived from brightness temperatures at 7 frequencies in the spin-rotation oxygen absorption band around 60 GHz, while humidity profiles from brightness temperatures at the same 5 frequencies used for IWV and LWP. Profiles were derived up to 10 km, with vertical resolution of 100 m up to 1 km and 250 m above. The retrieval algorithm is based on a back-propagation neural network regression algorithm trained on radiosoundings from both a coastal site, Brindisi Casale (WIGOS ID 0-20000-0-16320) that is the closest radiosounding station to Soverato located on the east coast of Italy, and from other radiosoundings at a mountain site (Solheim et al., 1998). The tipping curve calibration method (Han and Westwater, 2000) was used to ensure the accuracy of atmospheric water vapour measurements. This technique involves tilting the radiometer at different angles to observe the sky's microwave brightness temperature. This method allows for precise calibration by accounting for instrumental biases and environmental factors. Measurement uncertainties of MWPs for temperature profiling are discussed in Bock et al. (2024). Uncertainties are due to the instrument and to external sources, such as horizontal inhomogeneities of the atmosphere, topography, and interferences. Above the boundary layer, in the comparison with the radiosoundings, the retrieval uncertainty typically exceeds 1 K. For Relative Humidity (RH), the bias for a neural network retrieval, typically positive in particular in the free troposphere (Ware et al., 2003; Cimini et al., 2014; Xu et al., 2015; Caumont et al., 2016), is within 10-15% RH up to 7 km height, smaller above. The RMSE, estimated as the square root of the sum of the various contributors to the bias and the random error, is smaller than ~20% through the profile and in all weather conditions (Cadeddu et al., 2018). Uncertainties are not routinely evaluated per profile. These estimations do not account for the co-location uncertainty between radiosonde and the MWP.

The Infrared Thermometer (IRT) is a ground-based radiation pyrometer that measures the equivalent blackbody brightness temperature of the scene in its field of view. The temperature measuring range is from 173 to 473 K, with an accuracy of ±0.5 K + 0.7% of the temperature difference between the internal reference temperature and the object measured. Spectral sensitivity ranges from 9.6 to 11.7 μm.

The polarization Raman Lidar (Raymetrics LR111-D200) emits UV laser pulses at 355 nm with repetition frequency of 20 Hz and can detect the Raman backscattered radiation from atmospheric nitrogen at 387 nm, as well as parallel and cross polarized elastically backscattered radiation from atmospheric aerosol, clouds and molecules. Both Raman and elastic signals are acquired with raw vertical and time resolution of 7.5 m and 60 s, respectively. The calibration of polarization measurements was carried out using the method of ±45° rotation of a wave plate positioned in the lidar's optical path (Belegante et al., 2018). Lidar signals have been processed using EARLINET Single Calculus Chain (SCC), which allows for the data analysis of different lidar systems within EARLINET in an automated and unsupervised way (D'Amico et al., 2016; Mattis et al.,2016). Specifically, the vertical profiles of aerosol optical properties have been retrieved, namely aerosol backscattering and extinction coefficient (the latter for night-time measurements only), as well



as the particle linear depolarization ratio. The vertical range and effective resolution of these profiles are variable based on the signal-to-noise ratio of lidar detection channels, which in turn depends on the atmospheric scenario during measurements.

Additionally, a sun photometer (SP) was used to estimate the column aerosol optical depth (AOD) at different wavelengths, the column particle size distribution and precipitable water vapor provided by the AErosol RObotic NETwork program (AERONET). SP automatically tracks the sun and measures sunlight intensity at multiple wavelengths to determine aerosol properties (Boselli et al., 2012). Regular calibration ensures high accuracy. Lv2.0 data have been used in this paper. The network includes global stations with centralized data processing and quality control.

Data from the fifth-generation ECMWF atmospheric reanalysis (ERA5) of the global climate were used for comparison with observational measurements. ERA5 is the latest global climate reanalysis produced by ECMWF, providing hourly data on regular latitude-longitude grids at 0.25° x 0.25° resolution, with atmospheric parameters on 137 model levels interpolated to 37 pressure levels covering the period from 1940 to present. ERA5 is based on the Integrated Forecasting System (IFS) Cy41r2 release, which considered several significant improvements in the data assimilation methodology and representation of model processes for each component (atmosphere, ozone, land and ocean waves) compared to ECMWF's preceding reanalysis generation version, Cy31r12 of ERA-interim. Further, ERA5 combines a large variety of data sources, including more than 40 satellite sensors, such as AMSU-A and AMSU-B, IASI, and GNSS-RO, and uses advanced techniques to produce a coherent and detailed representation of past and present climate conditions. ERA5 performance has been evaluated in many studies for both surface and upper air levels at regional and global scales (e.g., Essa et al., 2022; Lavers et al., 2022; Jiao et al., 2021), and become one of the most frequently used datasets in climate applications and studies.

## 3   Measurements: analysis and results

The atmospheric circulation in summer 2021 was dominated by a strong 'blocking high' pressure system across southeastern Europe. This system initially expanded towards the east and was followed by another high-pressure system further west (ESOTC, 2021). These conditions favored dry weather and heatwaves with exceptionally high temperatures (Merlone et al., 2024), initially affecting the eastern and central Mediterranean areas and the Balkans, and then spreading to Spain, lasting until mid-August.

During the measurement period from June 24th to September 30th, the MP3014 observed frequent intermittent increases of water vapour content in the mid-troposphere, with high values of the water vapour mixing ratio (WVMR) and relative humidity (RH) in the altitude range between 650 and 450 hPa (Figures 2 and 3). The time series clearly shows the water vapour diurnal cycle along with intense moist structures periodically disappearing, with values larger than 6 g/kg ad often up to 100% RH. However, the MWR estimation of RH may be positively biased of about 10-15 % in the mid-troposphere. Intermediate periods were characterized by RH values below 70%. A decrease in the RH values in the mid-troposphere often corresponded to an increase in the values within the boundary layer. Freezing level was positioned at 4-5 km a.g.l., as derived from the MWP temperature retrievals. The relative humidity time series showed these high values on several days until August 10th, less frequently in the remainder of August and early September, and then again in late September. Low-pressure systems bringing clouds and rain, as well as a major issue to the measurements site power supply network, reduced and hindered reliable observations with the MWP during the full month of September. Very infrequent low clouds or short showers also affected a very minor fraction of the MWP observations (clearly visible in Figures 2 and 3 as profiles saturating the colour scale in a large vertical range).



It is worth noting that retrievals applied to microwave brightness temperatures at altitudes above 5 km provide a coarse resolution and may be influenced by the climatological values related to the input training datasets. This may introduce

significant errors in the retrieved values (Solheim et al., 1999). However, in terms of capturing the vertical gradient of humidity in the mid-troposphere, the neural network retrieval has already demonstrated to be quite efficient (Madonna et al., 2010). This is also due to the availability of the infrared sky temperature measured with the IRT, used to constrain the retrieval, indicating the presence of high concentration of water vapour or the presence of clouds in spectral window region. During the campaign period, the IRT measured persistent high temperature values of the clear sky, around 250-

260 K (Figure 4). These temperatures are consistent with the presence of a large amount of water vapour in the middle troposphere.

In Figures 2 and 3, the time series of the water vapour mixing ratio and relative humidity provided by ERA5, which is extracted for the nearest grid point to the Soverato site, are also shown. ERA5 shows relative humidity values around 500 hPa frequently between 60-80%, and sometimes larger than 90-95%, with a vertical structure in overall good agreement

with the MWR retrieval. Nonetheless, in the first half of July, located at upper-pressure range of 300-550 hPa there is, instead, not a good agreement between ERA5 and MWP measurements in both the depth and the time evolution of the middle-tropospheric moist layer. Moreover, the higher RH values estimated by ERA5 up to 300 hPa are not associated with the formation of cold clouds, neither in the ERA5 time series nor in the measurements collected with the laser ceilometer and the cloud radar operating during the campaign (Figures 6 and 7, respectively). This could be due to the

limited performance of ERA5 to correctly resolve the boundary layer at complex conditions. Limitations in this light have been also discussed in other studies, particularly during extreme events (e.g., Sinclair et al., 2022, Wei et al., 2024).

In terms of cloudiness, until July 2nd, ERA5 significantly overestimated cold clouds. A fair agreement between ERA5 and cloud radar observations regarding cold clouds was observed during the first half of the last week of June. However, in the second half of the same week, ERA5 continued to detect cold clouds above 6 km that were not observed by ground-

based remote sensing instruments.

The occurrence of heatwaves in long periods of the summer of 2021 likely inhibited strong convection and prevented the formation of clouds. This is supported by the balance of the CAPE (Convective Available Potential Energy) index, as provided by ERA5, shown in Figure 5. The CAPE index should assume values larger in the range 1000 - 2500 J/kg when moderate convection is active, while for strong convection, difficulty also generating stronger weather events, such as

sudden summer storms with wind gusts, values must be in the range 2500 - 4000 J/kg. Figure 5 shows that the CAPE index exceeds only sporadically 2500 J/Kg, excluding the occurrence of frequent strong convection, while, instead, moderate convection is more frequent from the second half of July. In a few periods, the values of CAPE are smaller, tending to a shallow or no convection, while for example at the beginning of the campaign, known as the period when in Sicily more than 48°C were reached (Merlone et al., 2024), the convection is moderate-to-strong.

Despite occasional low clouds in July and August, the laser ceilometer and cloud radar in Soverato did not detect overcast sky conditions, highlighting the value of combination of MWP/IRT measurements in detecting large concentrations of water vapour in the free troposphere. In August, ERA5 frequently detected clouds close to or at the top of the boundary layer, which were often not found in the observations, indicating the challenges of ERA5 in properly representing convection and warm cloud formation in the measurement campaign area. However, it must be highlighted measurements

at the Soverato site were conducted at a coastal location with a rising orography leaving the coast in a South-West direction. This geographic and meteorological setting adds complexity to forecasting cloud formation of both synoptic and convective nature within models. This complexity is also reflected in ERA5, especially in a region where satellite measurements are the primary source of support for the reanalysis data due to the absence of upper-air and ground-based remote sensing data. Moreover, it is worth mentioning that although the relatively high resolution of ERA5 compared to





the other global reanalyses, it is still on a scale that does not allow the model to resolve the convection process, but parametrise it. Additionally, it is known that IFS has inherent challenges in dealing with strong convection, particularly near coastlines (see ECMWF Forecast User Guide, section 9.6.1), as the case of the measurement site and pointed out in some studies (e.g., Lavers et al., 2022).

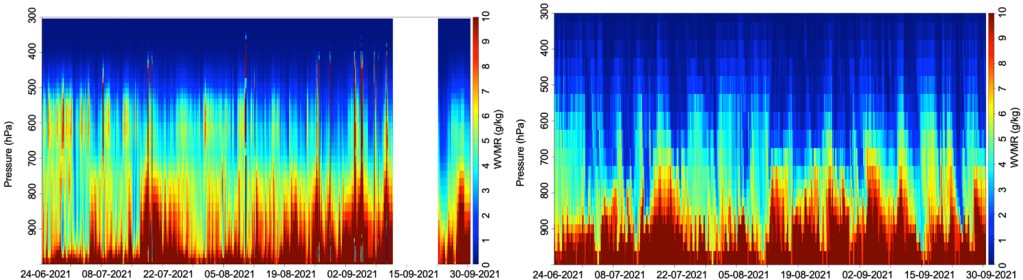

Figure 2: left panel, water vapour mixing ratio profiles from 1000 to 300 hPa as estimated using the microwave radiometer neural network retrieval from 24 June to 30 September 2021, with a time resolution of 5 minutes (last altitude level is set at 10 km which is lower than 300 hPa altitude level; right panel, same as left panel obtained using the ECWMF ERA5 reanalysis data, with a time resolution of 1 hour.

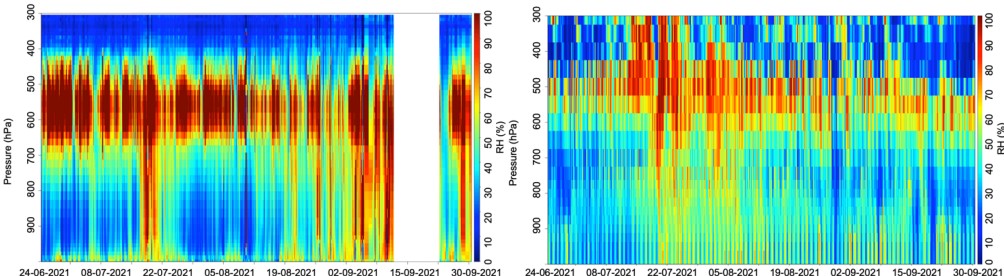

Figure 3: same as Figure 2, but for the corresponding values of relative humidity

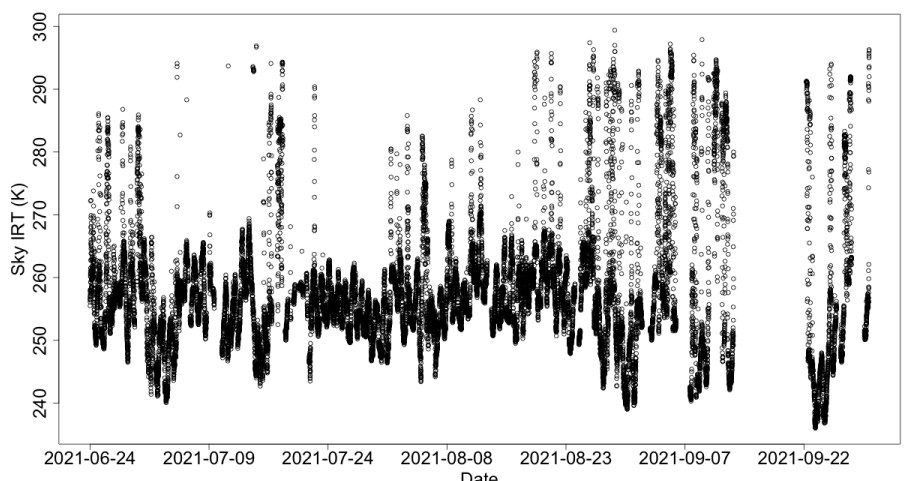





Figure 4: Infrared sky's temperature in the range of 9.6 to 11 microns as measured with the infrared thermometer, operating in synergy (same time sampling of 5 minutes) with microwave radiometer in Soverato.

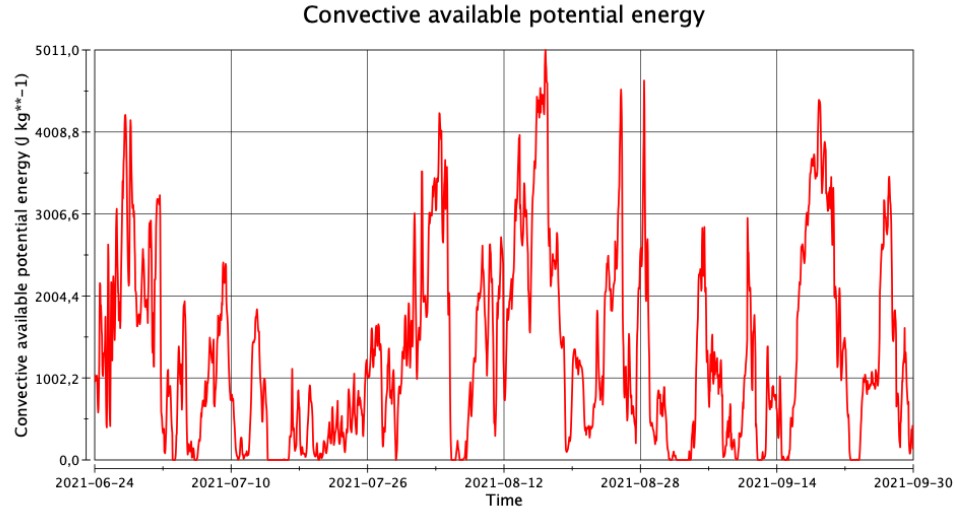


Figure 5: Convective Available Potential Energy (CAPE) retrieved from the ERA5 reanalysis data in the period 24 June to 30 September 2021.

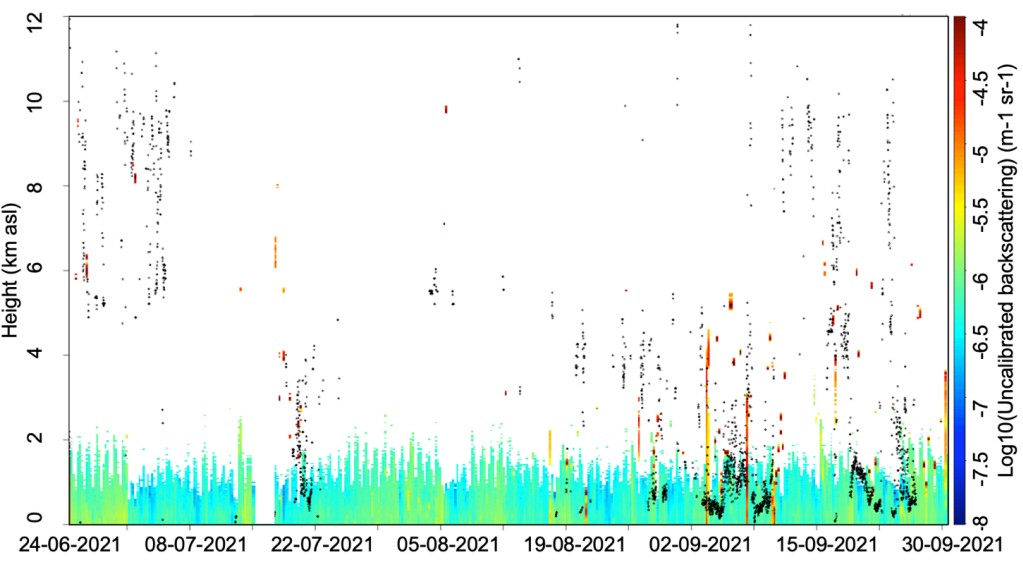





Figure 6: Uncalibrated aerosol backscattering coefficient (in dB) from the CL51 laser ceilometer, with corresponding cloud base height (black dots) from the ERA5 single level dataset for each of the closest grid points of the pixel containing the Soverato location.

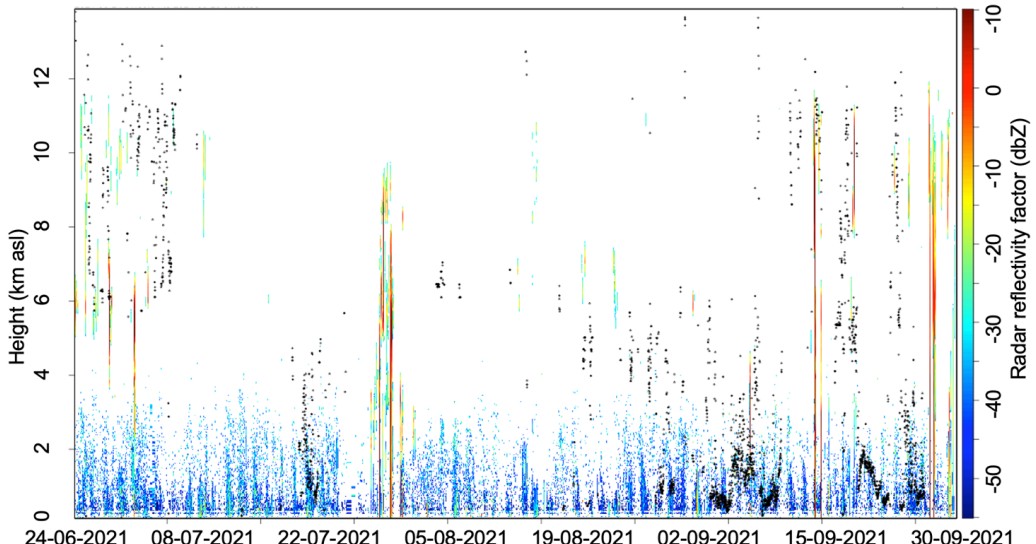

Figure 7: same as Figure 6, but for the equivalent radar reflectivity.

Previous studies showed good performances of the IFS scheme and the associated data assimilation in predicting the temperature field through comparison with radiosonde and satellite observations (Dyroff et al., 2015; Carminati et al., 2019) and an improved fit for temperature, wind and humidity in the troposphere in comparison with radiosonde data prior to assimilation (Hersbach et al., 2018). However, specific issues have been identified in the prediction and re-analysis of relative humidity in the upper troposphere and lower stratosphere, as well as with the general representation of ice supersaturation (Dyroff et al., 2015; Bland et al., 2021). In particular, in upper troposphere underestimations of

water vapour concentrations and ice supersaturation in ERA5 have been found (Kunz et al., 2014; Gierens et al., 2020; Schumann et al., 2021). Moreover, in comparison with homogenized datasets over the last four decades, also significant differences between radiosonde and reanalysis data have been found in the mid-upper troposphere depending on the latitude of the comparison (Madonna et al., 2022). ERA5 dry bias has been applied either using multiplication factors or

parameterized corrections, although proposed corrections do not consider spatial variations in the bias, particularly at different pressure levels (Wolf et al., 2023). It must be pointed out that the assessment of the model performance can vary with time due to system versioning and potential changes in the data assimilation or model schemes with the new releases. For instance, IFS used during the forecasting time of summer 2021(Cy47r2, implemented in May 2021) is different from the one implemented of 2016 to ERA5 (Cy41r2), and the present version (Cy48r1) which includes improvements to the

representation of moist physics in the model and increased satellite observation usage in cloudy regions in data assimilation, in a few months after the highlighted flood event (in Cy47r3 release, implemented in Oct. 2021).

An additional comparison to investigate the ERA5 dry bias is presented in Figure 8, which shows the error profile in ERA5 relative humidity (RH) with respect to the measurements by upper-air sounding balloons launched by the Potenza GRUAN station (WIGOS ID: 0-20008-0-POT; 40.60°N, 15.72°E, 760 m asl). GRUAN is a reference network that

provides traceable measurements with quantified uncertainties (Bodeker et al., 2016; Sommer et al., 2022). GRUAN data



products are not assimilated in ERA5 and, therefore, they are an independent reference comparator. Potenza is the only GRUAN station in Italy and is located in the southern part of the country, though it sits in the Apennine mountains, representing a much drier environment than Soverato. Nevertheless, the comparison with ERA5 data during June-September 2021 offers insight into the accuracy of ERA5 for RH.

During the campaign, the Potenza station performed one weekly launch, which is the minimum requirement for GRUAN. As a result, the comparison with ERA5 was based on a sample of 15 ascents. Figure 8 compares the bias and root-mean-square error of simultaneous RH profiles, using the ERA5 grid point nearest to Potenza. The observed bias ranges from -10% RH to +15% RH, from the near surface to the upper troposphere, while the root-mean-square error exceeds 30% RH across the entire profile. These results suggest that, even in a different environment, ERA5 exhibits a clear bias in RH

values and faces challenges in reproducing RH variability over time.

In the supplementary material, upper-air soundings from the Trapani Birgi RDS station (WIGOS ID: 0-20001-0-16429; 37.9142°N, 12.4914°E, 7 m asl) are shown alongside ERA5 hourly time series from the nearest reanalysis grid point. These soundings, performed twice daily (at 00 and 12 UTC) during the campaign period, provide further context. For Trapani station, the dry bias in ERA5 is generally smaller than that observed in Soverato, likely because ERA5 assimilates

the regular radiosonde data from Trapani.

Overall, the irregular behavior of RH error at different sites that has been shown draws attention to the possible need for caution when utilizing ERA5 RH data and suggests possible adjustments for further uses.

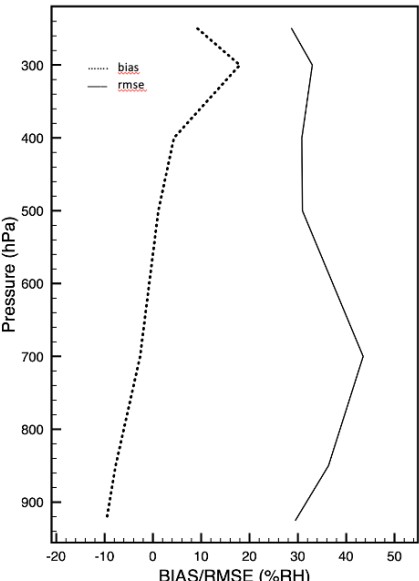

Figure 8: Bias and root-mean-square error for the ERA5 data compared to the upper-air measurements collected by the

GRUAN station in Potenza, Italy (WIGOS ID: 0-20008-0-POT; 40.60°N, 15.72E, 760 m asl), in the period June-September 2021.

The presented analysis of water vapour measurements available during MESSA-DIN is also supported by the precipitable water vapour (PWV) estimated by the MP3014 is compared in Figure 9 with retrievals obtained from the co-located sun photometer and the ERA5 estimate for the nearest grid point to Soverato. The time sampling of the three datasets differs:





hourly for ERA5, every 5 minutes for the MP3014, and approximately every 15 minutes for the sun photometer. ERA5
data is the smoothest compared to the other instruments, both of which have a small field of view.

During some periods, especially in the presence of thick clouds, sun photometer measurements are missing, as they are
automatically filtered during processing by AERONET. Some MP3014 values are affected by precipitation in the first
part of the campaign. During this period, when ERA5 overestimates the amount of clouds above 6 km, the MP3014 shows

a higher value of water vapour than the sun photometer and ERA5, consistent with a greater fraction of the total
atmospheric water being in the vapour phase.

Conversely, in the intermediate period of the campaign, the agreement between the MP3014 and the sun photometer is
good, with ERA5 generally showing the lowest values. In this period, ERA5 has intermediate values compared to the
other two instruments. It is important noting that, in addition to the pointed inconsistency in time resolution, ERA5 results

is an average value over a grid box of 31 km space, not a point value, which may contribute to the obtained underestimation
of the value due to sub-grid variability, particularly in convective situations. Overall, the three datasets tend to reproduce
comparable patterns. The sun photometer, which measures not vertically but along the direction of the sun, represents a
broader horizontal region of the atmosphere and it is sometimes in better agreement with ERA5, but most of the time is
closer to the PWV from the MWP.

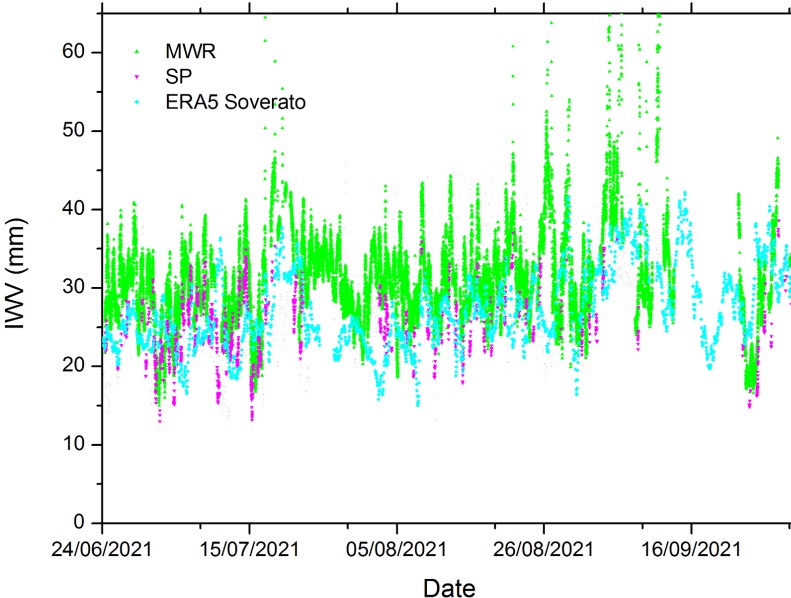


Figure 9: Integrated water vapour retrieved by the microwave radiometer (green), the sun photometer (magenta), and
obtained from the ECMWF ERA5 reanalysis (cyan) from 24 June to 30 September 2021 at Soverato measurement site.

To characterize the high RH values observed during the campaign in the mid-troposphere and their relationship with the
water vapour fluxes occurring over the Mediterranean basin, the ERA5 data have been used to estimate the total water

vapour transport (IVT), as the intensity of the IVT vector with the components: (Reynolds et al., 2022):



$$IVT = -\frac{1}{g}\int_{1000}^{300} vq\,dp$$

where g is the gravitational acceleration, v is the wind velocity, q is the specific humidity, p is the pressure, and the integration is from 1000 to 300 hPa. The period from June 24th to September 30th was characterised by high IVT values in different areas of the Mediterranean basis and involving also Soverato Gulf (plots every 2-hours for the first two

decades of July 2021 are provided in the Supplementary Information), mainly of provenance from NW, generated from the Spanish or French gulfs or the Tyrrenian sea, including the Soverato Gulf, or the Northern Africa coast. Figure 10 shows the IVT for the day of July 3rd over the Mediterranean and Central Europe, which reveals the south-eastern transport of a strong water vapour content with peak values over the Soverato gulf of 500 kg m⁻¹ s⁻¹. Figure 11 shows the corresponding synpotic scenario with the moist air mass flowing around the high pressure insisting in the western

Mediterranean and correlated with the occurrence of heatwave over South Italy (Wilgan et al., 2023). Another example is provided in Figure 12 for July 14th showing the transporter of similar content of water vapour from North Africa to South Italy boosting the low-pressure system over central Europe which generated flooding over west Germany. For both cases, the occurrence of high values of IVT over the measurement site corresponded to peaks in the values of relative humidity retrieved by the MP3014 at Soverato site. The investigation of the water vapour transport is made using ERA5

reanalysis data, despite the limitations discussed above, given its performance in terms of homogeneity and detection of the synoptic patterns. It must be pointed out that the comparison made between ERA5 and observations of the flooding that occurred in Germany revealed the good performance of the analysis in reproducing the patterns although the magnitude of event was underestimated (ECMWF, 2021).

Upon the comparisons shown in this paper, there are several reasons likely contribute to this underestimation; for example,

but not limited to, the bias in the RH values provided by the reanalysis; the underestimation of the IVT over the Mediterranean and more in general at the European scale, both in their magnitude and in their horizontal and vertical evolution in the atmosphere; also, the evaporation and convection parameterisation applied to the model microphysics have a key role in representing the amount of water vapour and clouds in the investigated domain. Furthermore, it is worth mentioning that it has been recognized that IFS versions released before the summer of 2022, such as in ERA5,

had suffered from a non-conservation water budget in their systems (Becker et al., 2022; Rackow et al., 2024), which potentially impacts the water vapour representation. Although the reported issue is considered to have a minor effect on the accuracy of numerical weather forecasts, it is acknowledged to be detrimental for the climate integrations where imprecise representation to the water vapour can affect the radiation energy budget of the atmosphere, ultimately leading to energy imbalance.



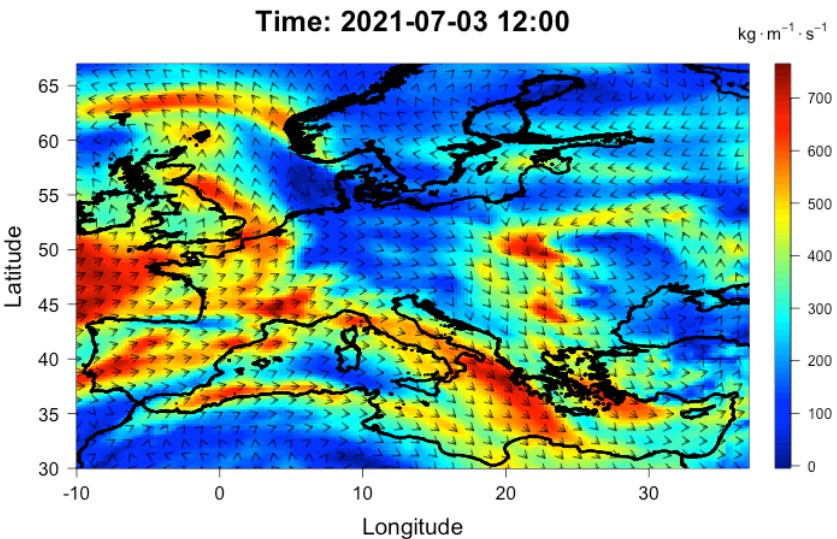


Figure 10: Map of the intensity and direction of the IVT vector on July 3rd over Europe and the Mediterranean basin.

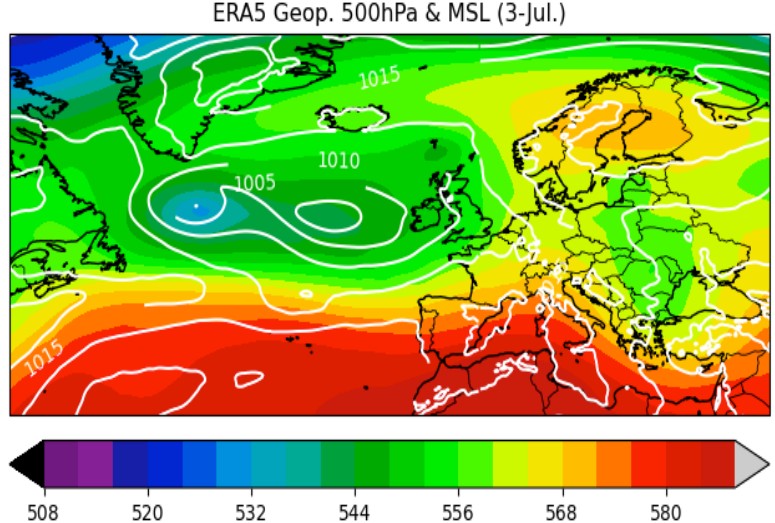

Figure 11: 500 hPa geopotential height and sea level pressure (contour lines) on July 3rd from ERA5 reanalysis at 12 UTC.



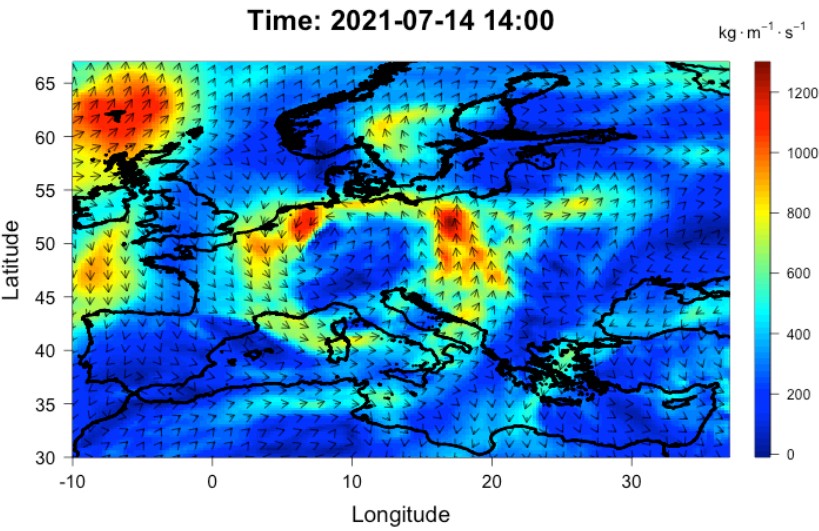


Figure 12: Same as Figure 10, but for the 14th July.

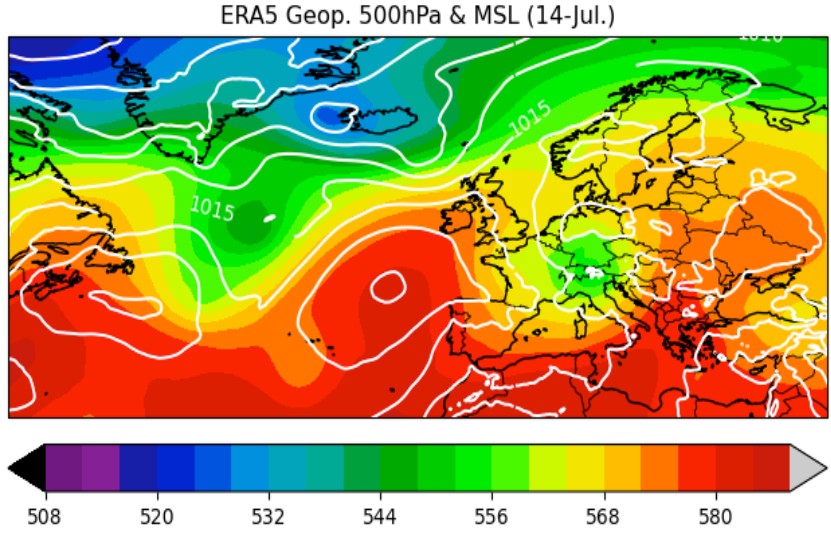

Figure 13: Same as Figure 11, but for July 14th at 12 UTC.

The increase in water content owing to anomalous fluxes in the Mediterranean, along with the frequently high aerosol concentration during the campaign period, did not increase cloud occurrence. The influence of aerosol types in cloud formation during the campaign was examined using UV Raman lidar measurements. To this purpose, Figure 14 shows values of the 355 nm lidar ratio of tropospheric aerosols, S, in the range between 1 and 6 km a.s.l. versus the corresponding





values of the particle depolarization ratio, δ, measured with the UV polarization Raman lidar during the campaign were investigated. It illustrates the large variability of S for all measurements collected during the campaign, mainly at night, during sessions averaging 2-3 hours. This variability highlights the different pure and mixed aerosol types observed on-site throughout the campaign. According to the literature (e.g. Müller et al., 2007; Burton et al., 2012; Groß et al., 2013; Papagiannopolous et al., 2018), the values of S and δ indicate the predominance of marine and continental aerosols, both pure and dust-contaminated, and a smaller amount of pure dust, which, from the range-resolved lidar measurements, is often located above 3 km. The analysis of the HYSPLIT air mass back trajectories reveals the main provenance from the Tyrrhenian and Adriatic seas for pure marine aerosol (trajectories can be retrieved from https://www.ready.noaa.gov/HYSPLIT.php at 1,3,and 5 km agl using the "Model vertical velocity" and a total run of 96 hours). Contribution from biomass burning aerosol was also present, mostly during the first part of the campaign, identified through photos and sky imagers' data collected at the site.

In terms of average aerosol size distribution, as retrieved using the SP, Table 1 presents a comparison between the volume concentrations and effective radius of aerosols estimated as monthly averages for July and September 2021 only because no inversion data from the SP data available in August 2021 (due to instrumental issues), hence no estimation of the aerosol size distribution parameters. Results reveal an increase in the volume concentration for the coarse fraction in September, although the dominant mode remains coarse, even in July. Properties of the fine particles are quite similar for both months, with a spreader distribution in September. However, for the coarse mode, the effective radius and volume concentration have a contrasting behaviour with the former larger in July and the latter larger in S. The dominance of the aerosol coarse fraction, in association with the evidence for the dominating contribution by marine aerosol and desert dust, implies that the coarse particles observed during the summer 2021 in Soverato, under atmospheric conditions, most of the time, dominated by high pressures insisting over South Italy, were another element decreasing the probability of occurrence of warm cloud formation at the measurement site. Despite the presence of ice nuclei at the site, of both mineral and marine nature, there was also a scarcity of high clouds during the campaign that, beyond the subsidence related to the high-pressure, was due to the occurrence of heat waves, often generating strong inversions in the temperature profiles acting as a barrier to vertical air movement, preventing the moist air at lower levels from rising to higher altitudes where cirrus clouds could form. The significant amount of water vapour in the mid-troposphere also amplifies the greenhouse effect and provides one of the most crucial amplifying feedback in the climate system (Dessler e a., 2008).





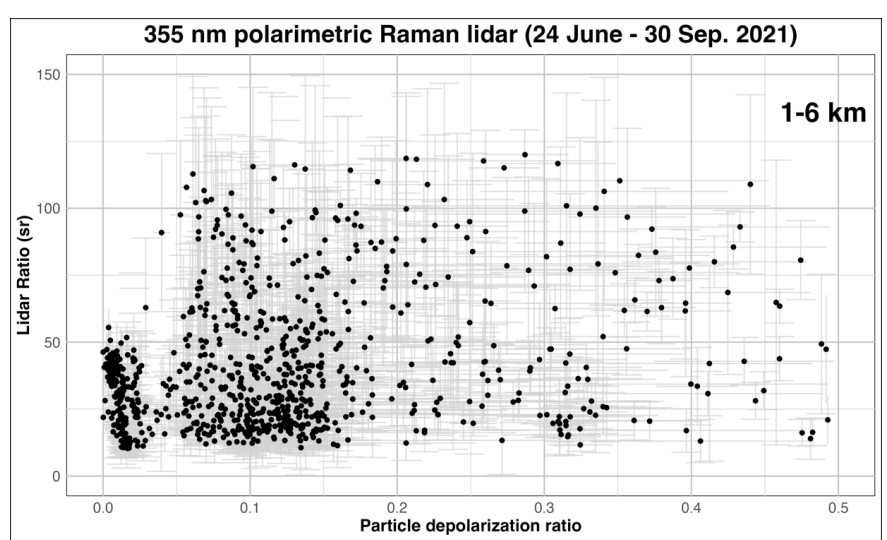

Figure 14: values of the 355 nm lidar ratio of tropospheric aerosols in the range between 1 and 6 km a.s.l. versus the corresponding values of the particle depolarization ratio measured with the UV polarization Raman lidar during the

campaign at Soverato from 24th June to 30th September 2021. The gray horizontal and vertical bars are representative of the corresponding statistical uncertainties of the plotted quantities. Only values with less than 40% relative uncertainty on the lidar ratios have been selected.

| Month | 2021-JUL | 2021-SEP |
|---|---|---|
| VolC-T | 0.10 | 0.15 |
| REff-T | 0.47 | 0.72 |
| VMR-T | 1.16 | 1.66 |
| Std-T | 1.32 | 1.06 |
| VolC-F | 0.02 | 0.02 |
| REff-F | 0.14 | 0.12 |
| VMR-F | 0.15 | 0.14 |
| Std-F | 0.47 | 0.57 |
| VolC-C | 0.08 | 0.14 |
| REff-C | 1.87 | 1.77 |
| VMR-C | 2.38 | 2.20 |
| Std-C | 0.68 | 0.65 |

Table 1: Monthly average values of the total volume concentration (VolC-T), fine mode concentration (VolC-F), and

coarse mode concentration (VolC-C) of the aerosol size distribution as estimated by the sun photometer during the campaign at Soverato for the months of July and September 2021. Reff, Std and VMR indicate the corresponding particle effective radius, the standard deviation of the effective radius and the volume mean radius. All the quantities are reported in μm. Data have been processed using the AERONET microphysical retrieval ( https://aeronet.gsfc.nasa.gov/new_web/Documents/Inversion_products_for_V3.pdf).




## 4    Discussion and conclusions

The MESSA-DIN campaign held at the coastal site of Soverato, in south Italy, during the Mediterranean summer of 2021 provided significant insights into the contribution of fluxes to the vertical concentration of water vapour in the troposphere. Observations showed frequent high relative humidity in the mid-troposphere during the campaign period boosted by water vapour fluxes correlated to the intense evaporation and the related water vapour fluxes.

The microwave radiometer (MP3014) and other instruments were used synergistically to provide a comprehensive understanding of water vapour profiles and cloudiness. Intense water vapour flux events were identified using the ERA5 data, primarily originating from the northwest Mediterranean (Spanish/French gulfs or Tyrrhenian Sea) and occasionally
from North Africa. These fluxes were often associated with high RH peaks at the measurement site, corresponding either to heatwaves or to significant synoptic patterns.

While ERA5 provided a coherent and detailed representation of synoptic patterns and showed a general agreement in the time evolution of the atmospheric vertical structure with observations, it exhibited a dry bias in RH values compared to the MWP. The magnitude of the bias is depending also on the bias affecting the MWP retrieval typically within 10-15 %
RH in the mid troposphere. For both the datasets, water vapour transport might be over or underestimated, and this may affect modeling of cloud formation in ERA5, which overestimated cold cloud presence, while ground instruments detected less frequent cloud cover, emphasizing the need for improving reanalysis performance in complex coastal and orographic settings. The bias in ERA5 was further assessed using GRUAN data from the Potenza station, which, through the comparison with SI traceable reference measurements, revealed a bias in the range from -10% RH to 15% RH. If ERA5,
a reanalysis product based on the Integrated Forecast System (IFS) model, does not accurately represent water vapour, then the parameterizations and physical processes related to water vapour in the implemented IFS version might need an upgrade.

Despite the coarse resolution in the free troposphere of Microwave Radiometers (MWP), the presented analysis confirms how atmospheric profilers can be extremely useful for validating and improving forecasts and modelling activities under
extreme or anomalous meteorological conditions. MWPs, for example, provide continuous measurements of temperature and water vapour in the atmosphere, and along with upper-air soundings and Raman lidars, both with a higher vertical resolution, can improve the knowledge of the water vapour fluxes, especially in areas like the Mediterranean Basin.  In the light of the resolution and forecast models, despite the computational challenges, having a km-scale forecast that allows better addressing for mesoscale phenomena, and resolving relevant processes like convection, instead of
parametrizing should be beneficial for such extreme events as proven in recent studies (Caldas-Alvarez et al., 2022; Fosser et al., 2024; Chang et al., 2024 ), and started to operational in some early warning systems and meteorological services such as the Limited Area Ensemble Prediction System developed by the COSMO consortium (COSMO-LEPS) and the German National Meteorological Service (DWD).

This study underscores the complexity of atmospheric processes in coastal regions and the need for multi-instrument
approaches and continuous validation of reanalysis products to improve weather and climate predictions. By improving the accuracy of water vapour flux predictions, we can better anticipate and mitigate the impacts of extreme weather events and climate variability in the Mediterranean. Persistent water vapour fluxes also impact the radiative balance, exacerbating the effect of surface radiation trapping, and potentially amplifying the effects of the heatwaves on human beings. Instead, large amounts of water vapour becoming available under these transport events can have profound impacts, particularly,
affecting Central and Eastern Europe. An example is the meteorological situation in Europe from 12 to 15 July 2021, characterized by a cutoff low-pressure system over Central Europe, supplying warm and very humid air from the



Mediterranean in its rotating movement. This low-pressure system led to heavy rainfall (more than 175 mm in 48 hours regionally), resulting in extensive flooding in Western Europe (Tradowsky et al., 2023). An incorrect prediction of moisture availability in the Mediterranean may become a critical factor in forecasting such extreme rain events.

We acknowledge that the analysis here relies on one specific case study based on the measurement availability which might introduce some uncertainty for general conclusions or imply different results with different atmospheric conditions and/or regions. For instance, Wu et al. (2024) investigated ERA5 performance in characterizing the pre-storm environment over China at different seasons and revealed that ERA5 tend to overestimate thermodynamic parameters, such as CAPE, convective inhibition, and Perceptible Water, while for dynamic parameters, such as vertical wind shear

and storm-relative helicity, it depicts an underestimation to them. Further work could aim at extending this study to include further cases, up on observational accessibility, and modelling experiments with a potential improvement to the data assimilation and models' configurations as discussed in this work.

**Data Availability Statement**

The data used in this study are generated by the Aerosol, Clouds and Trace Gases Research Infrastructure (ACTRIS) and
are available from the ACTRIS Data Centre using the following link: https://hdl.handle.net/21.12132/1.6882d49ed3ad4d4f. AERONET Data is available in near real-time on the AERONET website (https://aeronet.gsfc.nasa.gov/cgi-bin/data_display_aod_v3?site=Soverato_IMAA&nachal=2&level=2&place_code=10). The reanalysis data, ERA5, have been downloaded the Climate Date Store (CDS) of Copernicus Climate Change Service (C3S) which is publicly available
through https://cds.climate.copernicus.eu/cdsapp#!/home. The use of GRUAN data, available either from the CDS portal or the GRUAN website (www.gruan.org), with associated uncertainties is acknowledged.

**Author contributions**

FM conceived the study, designed the experiments, performed the synergic data analysis, and wrote the manuscript. BDR, DS, FM, MR, and SG carried out the measurement campaign with the local assistance of DM and MAM. All authors
contributed to the data analysis, and to the manuscript's review and editing.

**Competing interests**
The contact authors declare that they have no conflict of interest.

**Acknowledgements**

We acknowledge ACTRIS and Finnish Meteorological Institute for providing the data set which is available for download from https://cloudnet.fmi.fi. The access to the measurement site and the friendly logistic support by the Istituto Salesiano Sant'Antonio di Padova ("Salesiani Soverato") is gratefully acknowledged.

**Financial support**
This research has been partly supported by the ACTRIS-2 (grant no. 654109).

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
