# Peer review of "Water vapour fluxes at a Mediterranean coastal site during the summer of 2021: observations, comparison with atmospheric reanalysis, and implications for extreme events."

_Earth System Dynamics, 2024_

## Referee Comment (RC1)

**REVIEW of « Water vapour fluxes at a Mediterranean coastal site during the summer of 2021: observations, comparison with atmospheric reanalysis, and implications for extreme events. »**

This study describes the atmospheric measurements during the campaign MESSA-DIN during summer 2021, in particular comparing different observational data to ERA5 reanalysis estimates. The study investigates the differences between the two types of data, in the setting of previous literature assessing ERA5 performance. However, the manuscript lacks in preciseness and often omits the link between the data and scientific assertions. I am willing to recommend publication of this work only after the language and scientific statements are reviewed across the whole text. The latter should fit with  the contents of the study.

**GENERAL COMMENTS**
- In the « Instruments and datasets » section please remove focus on the technicalities in favour of explaining more clearly how the measurements are to be interpreted, referring to the specific plots (e.g., how you infer presence of cloud cover).
- What do you mean by water vapour fluxes ? Are you talking about atmospheric transport or of fluxes from the surface to the atmosphere ? Please clarify in the whole text, as this is one of the main points you touch in the Introduction and Conclusions.
- In the conclusions you should acknowledge more clearly that the comparison with reanalysis is hindered by its coarse resolution compared to station measurements. In general, you also say that this highlights the need of better representations of coastal / orographic atmospheric processes (e.g. line 416). However, this is possible only with altogether different high-resolution models. We cannot expect a ~31 km resolution model to contain the topographic details that allow an improved comparison with your data.

**TECHNICAL COMMENTS**
The technical comments touch the problems of language and precision, which should be addressed more generally than in the individual comments, as indicated in the introductory paragraph of the review.

Line 35 : feeding the zonal or meridional air mass transport with what ? Representing a significant source of what ?
Line 37 : Are you saying that both remote water vapour sources and local moisture uptakes from intense evaporation are necessary for severe precipitation ? This is not clear in the sentence.
Line 41 : substitute « while » with « and »
Line 46 : remove « among the last decades »
Line 50 : remove « and recent studies » if you are not including additional references.
Line 52 : I suggest to reformulate with « around or higher than the climatological values ».
Line 55 : Expand acronym CIAO.
Line 57 : « at a coastal site the effect of... ».
Line 64 : « collected with a microwave radiometer and an infrared thermometer over 24h/7days from June 24 to September 30 2021. »
Line 65 : « In the time series, high values of relative humidity in the mid-troposphere were investigated… ». Note also repetition of term « contribution ».
Line 68 : What do you mean by « local paucity of warm and cold cloud layers » ?
Line 73 : « provides » to replace « offers ».
Line 75 : « the investigation of the rôle of aerosols... »
Line 88 : remove « presented in this paper ».
Line 102 : expand acronym SNR.
Line 175 : I do not agree with the sentence « A decrease in the RH values in the mid-troposphere

often corresponded to an increase in the values within the boundary layer. ». In my opinion there is no clear pattern between RH in the mid troposphere and in the boundary layer, as BL-maxima seem to occur both with mid-tropospheric minima and maxima.

Line 177 : What levels are you talking about ?

Line 195-197 : The sentence is badly structured. Please correct.

Line 200 : replace « in this light » with « Limitations of this kind ».

Line 202 : Add reference to figure in the first sentence in paragraph.

Line 210-210 : Convection is linked to the presence of strong CAPE, but also depends on low values of convective inhibition (CIN) or on the presence of forcing overcoming CIN. For this reason you cannot directly relate CAPE and convection. In these sentences, you should moderate the strength of your assertions.

Lines 215-217 : Insert ref to figure.

Line 252-253 : I don't understand what you mean by « improved fit for temperature, wind and humidity in the troposphere  in comparison with radiosonde data prior to assimilation ». Are you comparing the IFS performance with radiosonde data before and after data assimilation ?  Please clarify.

Line 255-256 : please rephrase to « in the upper troposphere ERA5 is found to underestimate water vapour concentration and ice supersaturation ».

Line 259 : « addressed » to replace « applied ».

Line 266 : the reference to the flood event is out of place here. If possible, remove mention of the Cy47r3 release. Or else clarify differences with respect to other versions in a separate sentence.

Line 271 : remove « comparator ».

Line 272 : separate the two sentences. Note that the second is not in opposition with the first, as the term « though » would suggest.

Line 279 : « even in a different environment than … NAME FIRST ENVIRONMENT »

Line 293 : « MP3014, WHICH is compared in …  ».

Line 297 : Merge with previous paragraph. Also, in this and in the following paragraph, add ref to Figure 9.

Line 315 : Not clear what you mean by « is the intensity of the IVT vector with the components ».

Line 217 : if v is vector denote with vector symbol (arrow or bold).

Line 320 : note that decade means ten years. Please correct.

Line 319-321 : The sentence from « involving » to « Africa coast » is confused. Please separate from previous sentence and rephrase to make it comprehensible.

Line 323 : Please add label on plot corresponding to the position of Soverato.

Line 234 : « transport ».

Line 235 : refer to Figure 13.

Line 334 : « contributing ».

Line 334-338 : replace « ; » with full stops. Also, please rephrase and create logical connections between the sentences. At the moment it is difficult to follow.

Line 341-344 : Only the first part of the sentence is relevant for this study. Please remove misleading reference to climate integrations.

Line 371-373 : Rephrase and separate sentence starting from « only because ».

Line 375 : replace « spreader » with « wider ».

Line 375-376 : sentence is not clear.

Line 376-380 : The whole period is confused, and it is not clear on which data/analysis you base your statement. Please improve clarity and precision.

Line 385 : Since you have no direct measurement of the water-vapour greenhouse effect impact on temperature, you can cite this as a potentially amplifying factor.

Line 408-410 : You should be more specific about the number of case studies. I believe you analyse water vapour transport rather than fluxes from the sea. Say « Two intense water vapour **transport** events were identified using the ERA5 data, one from the northwest Mediterranean (Spanish/French gulfs or Tyrrhenian Sea) and one from North Africa ».

Line 416 : « compared to ground instruments detecting less frequent cloud cover. This emphasizes the need for improving reanalysis performance in complex coastal and orographic settings. »

Line 428-433 : The whole period is impossible to follow. Please rephrase and separate in multiple sentences to make it clear.

Line 440 : « affecting Central and Eastern Europe » Can you say this based on climatological studies ? If not, please specify you are talking of one example and do not generalise.

Line 451-452 : The last sentence is badly structured. Please revise.

---

## Referee Comment (RC2)

**A review of Earth System Dynamics.**

**MS title:**

**"Water vapour fluxes at a Mediterranean coastal site during the summer of 2021: observations, comparison with atmospheric reanalysis, and implications for extreme events"**

Madonna et al, 2024 an observational-core study that explores the use of high-quality observations and measurements of hydrological atmospheric features in a specific location where such measurements are deemed crucial for weather and climate predictability. They compare the results with global ECMWF ERA-5 reanalysis in the roughly corresponding grid points, to highlight the large deviations between one of the most highly used meteorological tools and observations. An emphasis is given to the role of integrated water vapour transport in generating extreme weather events, and the importance of correctly evaluating these measurements in reanalysis.

The objective of improving the performance of reanalysis data using observations is important, especially in coastal areas that are affected by sub-grid processes, and the results presented in this MS are undoubtedly of interest to the scientific community. Indeed, it is evident that major efforts were invested in this campaign, and for good reasons.

However, I suggest a major revision of the MS, especially concerning the present structure, but also possibly the choices made by the authors:

**Major comments:**

1) The choice of comparing observations to a relatively coarse reanalysis is unclear. Would it not be more beneficial to compare against a higher-resolution alternative?

2) In the introduction, the authors elaborate on the methods used at a highly technical level, while not emphasizing enough the motivation and knowledge gaps addressed by the study, and how the observations serve those objectives. I suggest moving the technical details to an appendix and expanding the introduction on the role of moisture fluxes in the atmosphere, why they are misrepresented especially in coastal areas in the Mediterranean, the implications, and the importance of the detailed observations you present for their better prediction.

3) The title of the MS could be more specific, possibly naming the location or the campaign itself. Even the scope of the results discussed throughout is larger than suggested by the title. Furthermore,

the discussion of the water-vapor fluxes is brief and does not provide new insights as for the well-known importance of these fluxes for extreme weather events. The title suggests that the water vapor fluxes were derived from observation, but that's not the case. Therefore, I suggest rephrasing the title to describe the MS more accurately.

4) Some of the figures (4, 6, and 14) show raw data, with highly technical captions that are not intuitive for non-expert readers. Seeing that the focus of ESD is usually not set to such levels of technicality, I recommend adding a line to the captions that recalls the physical meaning of the observed quantity.

5) The summary should highlight the importance of the field campaign: what new information was gained and how can it be harnessed to improve our understanding or the model performance? Simply pointing out the biases seems like an underreaching conclusion, especially when comparing measurements to global reanalysis data that does not even have a corresponding grid point in the location of interest.

**Minor comments:**

L65: "investigated to identify the contribution by the contribution of water

vapour fluxes and convection" – unclear

Fig.1: Consider adding a larger view pointing out the location of Soverato for geographical orientation.

L88: instruments

L127: **and** smaller above.

L148-149: this sentence seems interrupted

L173: ad→and

L174: biased **at**

L208: larger values

L209: rephrase. The sentence is unclear.

L219: highlighted **that** measurements

L250-265: This fits better in the introduction

L304: noting→ to note

L324: insisting→ persisting

L325: of **a** heat wave

L326: transporter→ transport?

L334: contributing or that likely contribute.

L343: representation **of**

L376-380: long sentence, consider splitting.

L428-433: long sentence with several errors. Please rephrase.

---

## Author Comment (AC1)

Reply to the Anonymous Referee #1

**REVIEW of « Water vapour fluxes at a Mediterranean coastal site during the summer of 2021: observations, comparison with atmospheric reanalysis, and implications for extreme events. »**
This study describes the atmospheric measurements during the campaign MESSA-DIN during summer 2021, in particular comparing different observational data to ERA5 reanalysis estimates. The study investigates the differences between the two types of data, in the setting of previous literature assessing ERA5 performance. However, the manuscript lacks in preciseness and often omits the link between the data and scientific assertions. I am willing to recommend the publication of this work only after the language and scientific statements are reviewed across the whole text. The latter should fit with the contents of the study.

**Thank you for your constructive feedback on our manuscript. We appreciate your valuable observations, which have highlighted areas for improvement and the need to strengthen the connection between our data and the scientific assertions. Below, we provide (in bold) a detailed response to each of the reviewer's comments.**

**GENERAL COMMENTS**
- In the « Instruments and datasets » section please remove focus on the technicalities in favour of explaining more clearly how the measurements are to be interpreted, referring to the specific plots (e.g., how you infer presence of cloud cover).

**Section 2 has been shortened and revised to achieve a better balance between providing sufficient technical details to ensure confidence in the measurements discussed in the manuscript and improving the clarity of their interpretation as presented in the figures.**
**The new version of Section 2 is reported below:**

**"The study employs a suite of advanced instruments for studying aerosol, water vapour, and clouds. The measurement site was equipped with several instruments, including a Ka-band Doppler radar, laser ceilometer, a UV polarization Raman lidar, a microwave radiometer complemented by an infrared radiometer, a sun photometer, a wind Doppler lidar, a total sky imager, and other near-surface measurements. This section describes the instruments and products used in the data analysis presented in this paper.**

**The CL51 ceilometer measured cloud base height and attenuated backscattering up to 15 km a.g.l. with high temporal (10 s) and spatial (10 m) resolutions using a 910 nm laser diode system, though limited by a low signal-to-noise ratio (Madonna et al., 2018). Data were processed via the ACTRIS-Cloudnet algorithm (Rosoldi, 2024) and provide the uncalibrated backscattering coefficient, which depends on the amount and size of particles at different altitude levels. Except for very thin clouds or huge aerosol outbreaks, this allows distinguishing between clouds and aerosol particles depending on the backscattering intensity.**

**The MIRA-36 Ka-band radar, designed for unattended long-term operation, provided range-resolved cloud reflectivity, vertical wind, and linear depolarization ratio up to 15 km with 30 m spatial and 30 s temporal resolutions. The radar is also crucial for cloud and large aerosol detection (Madonna et al., 2010, 2013). In this paper, the radar reflectivity factor, proportional to the sixth power of the observed particle size, is used to detect clouds and precipitation, related, in**

the measurement campaign, to values larger than -30 dBZ, while values lower than -40 dBZ are referred to echoes from aerosols, including pollen or insects.

A Doppler lidar operating at 1.5 μm measured backscattered radiation and radial velocity up to 12 km, depending on atmospheric aerosol load, with a Doppler precision below 20 cm/s for SNR > -17 dB. Vertical wind profiles were retrieved using conical scans every 5 minutes, following Päschke et al. (2015).

The Radiometrics Microwave Profiler (MWP 3014) measured key atmospheric parameters, including integrated water vapor (IWV), liquid water path (LWP), brightness temperatures, and thermodynamic profiles (Cimini et al., 2018; Madonna et al., 2010). Measurements continued until a power supply failure on September 30. IWV and LWP were derived from brightness temperatures at 22.235–30.000 GHz, while temperature and humidity profiles were retrieved using neural networks trained on coastal and mountain radiosonde data (Solheim et al., 1998). Vertical resolutions reached 100 m below 1 km and 250 m above. Calibration was ensured via the tipping curve method, mitigating instrumental biases (Han and Westwater, 2000). Despite robust calibration, uncertainties were notable: temperature retrieval errors exceeded 1 K above the boundary layer (Bock et al., 2024), and relative humidity retrievals showed biases within 10–15% up to 7 km (Ware et al., 2003; Cimini et al., 2014; Xu et al., 2015; Caumont et al., 2016). RMSE for RH was below ~20% across profiles under all conditions (Cadeddu et al., 2018), although co-location issues between radiosonde and MWP measurements could contribute to further errors. These factors highlight the importance of refining retrieval methods for improved accuracy.

A polarization Raman lidar (Raymetrics LR111-D200) captured aerosol and cloud properties using 355 nm UV pulses, with processed data providing backscattering, extinction coefficients, and depolarization ratios via EARLINET's SCC (D'Amico et al., 2016).

Sun photometer (AERONET Lv2.0) measurements offered aerosol optical depth (AOD), particle size distribution, and water vapor estimates with stringent calibration and quality control (Boselli et al., 2012).

For comparison, the ERA5 reanalysis dataset, produced with ECMWF's Integrated Forecasting System Cy41r2, provided hourly atmospheric parameters at a 0.25° resolution, incorporating data from over 40 satellite systems (Hersbach et al., 2020; Essa et al., 2022; Lavers et al., 2022). Further, ERA5 combines a large variety of data sources, including more than 40 satellite sensors, such as AMSU-A and AMSU-B, IASI, and GNSS-RO, and uses advanced techniques to produce a coherent and detailed representation of past and present climate conditions. Despite its widespread use, ERA5 requires validation against high-resolution observational data to improve its applicability in sub-grid process-sensitive coastal areas."

- What do you mean by water vapour fluxes ? Are you talking about atmospheric transport or of fluxes from the surface to the atmosphere ? Please clarify in the whole text, as this is one of the main points you touch in the Introduction and Conclusions.

In the entire paper, we are referring to enhanced water vapour transport in the free troposphere due to anomalous evaporation from the sea surface, enhancing other remote sources. The updated version of the manuscript will be entirely revised to reflect this concept.

- In the conclusions you should acknowledge more clearly that the comparison with reanalysis is hindered by its coarse resolution compared to station measurements. In general, you also say that this highlights the need of better representations of coastal / orographic atmospheric processes (e.g. line 416). However, this is possible only with altogether different high-resolution models. We cannot expect a ~31 km resolution model to contain the topographic details that allow an improved comparison with your data.

**The sentence at lines 415-418 has been rephrased as follows: "For both datasets, water vapour transport may be either overestimated or underestimated, potentially affecting the modeling of cloud formation in ERA5. ERA5 overestimated cold cloud presence, whereas ground-based instruments detected less frequent cloud cover. This discrepancy highlights the need for higher-resolution (regional) reanalysis with improved performance in capturing water vapour variability in complex coastal and orographic settings.". The assessment of the performance of regional reanalysis datasets, such as CERRA, does not show notable improvements for water vapour, either at the surface or in the free troposphere, in contrast to temperature and wind. For instance, see the comparative performance analysis of CERRA and ERA5 (Ridal et al., 2024, QJRMS). Furthermore, CERRA data are not yet officially available for the entire campaign period (up to June 2021). Moreover, the nearest grid point of ERA5 is at a distance from Soverato of about 7.8 km — Soverato (38.6894°N, 16.5453°E) and the ERA5 point (38.75°N, 16.5°E) — both over land. Given that we are not reporting single comparison profiles but a time series of more than 3 months and the variability of the water vapour field, it is unlikely that there is a significant bias related to the spatial mismatch of the two datasets. Additional details are provided in the response to the third general comment from anonymous reviewer #2.**

TECHNICAL COMMENTS
The technical comments touch the problems of language and precision, which should be addressed more generally than in the individual comments, as indicated in the introductory paragraph of the review.
Line 35 : feeding the zonal or meridional air mass transport with what ? Representing a significant source of what ?
**The sentence has been rephrased as: "The Mediterranean summer is often characterized by significant water vapor fluxes from the sea, driven by intense evaporation, which are a significant source of  moisture for both zonal and meridional air masses."**
Line 37: Are you saying that both remote water vapour sources and local moisture uptakes from intense evaporation are necessary for severe precipitation ? This is not clear in the sentence.
**The sentence has been modified as follows as: "Severe precipitation in the Mediterranean Basin depends on both remote and local sources of anomalously intense surface evaporation"**
Line 41 : substitute « while » with « and »
**OK**
Line 46 : remove « among the last decades »
**OK**
Line 50 : remove « and recent studies » if you are not including additional references.
**OK**
Line 52 : I suggest to reformulate with « around or higher than the climatological values ».
**OK**
Line 55 : Expand acronym CIAO.

**OK**, CNR-IMAA Atmospheric Observatory

Line 57 : « at a coastal site the effect of... ».

**OK**

Line 64 : « collected with a microwave radiometer and an infrared thermometer over 24h/7days from June 24 to September 30 2021. »

**OK**

Line 65 : « In the time series, high values of relative humidity in the mid-troposphere were investigated... ». Note also repetition of term « contribution ».

**OK, this will be fixed.**

Line 68 : What do you mean by « local paucity of warm and cold cloud layers » ?

The sentence has been modified as follows

Line 73 : « provides » to replace « offers ».

**OK**

Line 75 : « the investigation of the role of aerosols... »

**OK**

Line 88 : remove « presented in this paper ».

**OK**

Line 102 : expand acronym SNR.

**OK, Signal-to-Noise Ratio.**

Line 175 : I do not agree with the sentence « A decrease in the RH values in the mid-troposphere often corresponded to an increase in the values within the boundary layer. ». In my opinion there is no clear pattern between RH in the mid troposphere and in the boundary layer, as BL-maxima seem to occur both with mid-tropospheric minima and maxima.

**This sentence, as not essential to convey the most important message of the data analysis, has been removed.**

Line 177 : What levels are you talking about ?

**At this line, we just find the word "freezing level". Not clear the reviewer's comment; however, that part of the manuscript will be revised.**

Line 195-197 : The sentence is badly structured. Please correct.

**The sentence has been rephrased as follows: "In the first half of July there is, instead, not a good agreement between ERA5 and MWP measurements in range 300 hPa-550 hPa, for both the depth and the time evolution of the middle-tropospheric moist layer.**

Line 200 : replace « in this light » with « Limitations of this kind ».

**OK**.

Line 202 : Add reference to figure in the first sentence in paragraph.

**OK**

Line 210-210 : Convection is linked to the presence of strong CAPE, but also depends on low values of convective inhibition (CIN) or on the presence of forcing overcoming CIN. For this reason you cannot directly relate CAPE and convection. In these sentences, you should moderate the strength of your assertions.

**In the attached plot, the CIN for the Soverato site in the period June-September 2021 is reported. We will moderate the tone of the sentence, although it is evident that, for long periods of time (e.g. in June-July) the CIN values is frequently zero, but the convention is often (July) low-moderate.**

[Figure]

Data Min = 0,3, Max = 993,9

Lines 215-217 : Insert ref to figure.

**OK**

Line 252-253 : I don't understand what you mean by « improved fit for temperature, wind and humidity in the troposphere in comparison with radiosonde data prior to assimilation ». Are you comparing the IFS performance with radiosonde data before and after data assimilation ? Please clarify.

**The sentence refers faithfully to Hersbach et al., 2020, which introduce and assesses ERA5 performances. This will be clarified in the text.**

Line 255-256 : please rephrase to « in the upper troposphere ERA5 is found to underestimate water vapour concentration and ice supersaturation ».

**OK**

Line 259 : « addressed » to replace « applied »

**OK**

Line 266 : the reference to the flood event is out of place here. If possible, remove mention of the Cy47r3 release. Or else clarify differences with respect to other versions in a separate sentence.

**OK, the difference with respect to other versions will be clarified.**

Line 271 : remove « comparator ».

**OK**

Line 272 : separate the two sentences. Note that the second is not in opposition with the first, as the term « though » would suggest.

**OK.**

Line 279 : « even in a different environment than … NAME FIRST ENVIRONMENT »

**The sentence will be modified as follows: "These results suggest that, even in a mountain environment, ERA5 exhibits a clear bias in RH values and faces challenges in reproducing RH variability over time."**

Line 293 : « MP3014, WHICH is compared in … ».

**OK**

Line 297 : Merge with previous paragraph. Also, in this and in the following paragraph, add ref to Figure 9.

**OK**

Line 315 : Not clear what you mean by « is the intensity of the IVT vector with the components

**The sentence will be modified as follows: "... the ERA5 data have been used to estimate the total water vapour transport (IVT) and its zonal and meridional components."**

Line 217 : if v is vector denote with vector symbol (arrow or bold).

**OK.**

Line 320 : note that decade means ten years. Please correct.

**OK.**

Line 319-321 : The sentence from « involving » to « Africa coast » is confused. Please separate from previous sentence and rephrase to make it comprehensible.

**OK**.

Line 323 : Please add label on plot corresponding to the position of Soverato.

**OK.**

Line 234 : « transport »

**OK**.

Line 235 : refer to Figure 13.

**OK**

Line 334 : « contributing ».

**OK**

Line 334-338 : replace « ; » with full stops. Also, please rephrase and create logical connections between the sentences. At the moment it is difficult to follow.

Line 341-344 : Only the first part of the sentence is relevant for this study. Please remove misleading reference to climate integrations.

Line 371-373 : Rephrase and separate sentence starting from « only because ».

Line 375 : replace « spreader » with « wider ».

**OK**

Line 375-376 : sentence is not clear.

**The sentence has been rephrased as follows: "Properties of the fine particles are quite similar for both months, with a spreader distribution in September, except for the coarse mode.**

Line 376-380 : The whole period is confused, and it is not clear on which data/analysis you base your statement. Please improve clarity and precision.

**Summer 2021 has been replaced with Jun-Sep. 2021,**

Line 385 : Since you have no direct measurement of the water-vapour greenhouse effect impact on temperature, you can cite this as a potentially amplifying factor.

**OK, the sentence will include a "may" to mitigate the statement.**

Line 408-410 : You should be more specific about the number of case studies. I believe you analyse water vapour transport rather than fluxes from the sea. Say « Two intense water vapour **transport** events were identified using the ERA5 data, one from the northwest Mediterranean (Spanish/French gulfs or Tyrrhenian Sea) and one from North Africa ».

**OK**

Line 416 : « compared to ground instruments detecting less frequent cloud cover. This emphasizes the need for improving reanalysis performance in complex coastal and orographic settings. »

**See the replies to the general comments.**

Line 428-433 : The whole period is impossible to follow. Please rephrase and separate in multiple sentences to make it clear.

**The sentence has been rephrased as follows: "In light of the resolution and forecast models, having a km-scale forecast can significantly improve the ability to address mesoscale phenomena. Despite the computational challenges, this approach allows for the resolution of important processes such as convection, rather than relying on parameterization. Recent studies (Caldas-Alvarez et al., 2022; Fosser et al., 2024; Chang et al., 2024) have shown that this is beneficial for extreme events. Furthermore, such models have already been implemented in some early warning systems and meteorological services, including the Limited Area Ensemble Prediction System (COSMO-LEPS) developed by the COSMO consortium and the German National Meteorological Service (DWD)."**

Line 440 : « affecting Central and Eastern Europe » Can you say this based on climatological studies ? If not, please specify you are talking of one example and do not generalise.

**The sentence has been rephrased as follows: "Instead, according to the investigated cases, large amounts of water vapour becoming available under these transport events may have profound impacts, particularly, affecting Central and Eastern Europe."**

Line 451-452 : The last sentence is badly structured. Please revise.

**The sentence has been rephrased as follows: "Future work could focus on expanding this study to include additional cases, depending on observational availability, as well as conducting modeling experiments to improve data assimilation and model configurations, as discussed in this work."**

---

## Author Comment (AC2)

Reply to the Anonymous Referee #2

A review of Earth System Dynamics. MS title:
"Water vapour fluxes at a Mediterranean coastal site during the summer of 2021: observations, comparison with atmospheric reanalysis, and implications for extreme events"
Madonna et al, 2024 an observational-core study that explores the use of high-quality observations and measurements of hydrological atmospheric features in a specific location where such measurements are deemed crucial for weather and climate predictability. They compare the results with global ECMWF ERA- 5 reanalysis in the roughly corresponding grid points, to highlight the large deviations between one of the most highly used meteorological tools and observations. An emphasis is given to the role of integrated water vapour transport in generating extreme weather events, and the importance of correctly evaluating these measurements in reanalysis.
The objective of improving the performance of reanalysis data using observations is important, especially in coastal areas that are affected by sub-grid processes, and the results presented in this MS are undoubtedly of interest to the scientific community. Indeed, it is evident that major efforts were invested in this campaign, and for good reasons.
However, I suggest a major revision of the MS, especially concerning the present structure, but also possibly the choices made by the authors:

**Thank you for your detailed review and valuable feedback on our manuscript. We are glad to see that you recognize the importance of the study's objectives. We also appreciate your acknowledgment of the effort invested in the observational campaign and the relevance of the results to the scientific community. Below, we provide our responses (in bold) to each of the reviewer's comments.**

Major comments:
1)  The choice of comparing observations to a relatively coarse reanalysis is unclear. Would it not be more beneficial to compare against a higher-resolution alternative?
**Theoretically, we agree with the reviewer. We also considered the idea of comparing with CERRA, the ECMWF regional reanalysis, but currently, data in the CDS are available only until June 2021. During the last C3S GA, we interacted with the ECMWF team producing CERRA, who offered updated datasets until 2024. However, they clarified that for RH, the advantage of using CERRA compared to ERA5 is quite limited. The recent paper by Ridal et al. (2024, QJRMS) confirms this outcome. Nevertheless, the manuscript text at lines 415-418, as suggested by anonymous reviewer #1, will be amended to properly discuss the limitations of a higher-resolution reanalysis in capturing orographic effects. In follow-up activities, the authors also plan to compare the data from the campaign with a downscaled ERA5 version using explicit convection parameterization or to carry out a comparison with the WRF model**

2)  In the introduction, the authors elaborate on the methods used at a highly technical level, while not emphasizing enough the motivation and knowledge gaps addressed by the study, and how the observations serve those objectives. I suggest moving the technical details to an appendix and expanding the introduction on the role of moisture fluxes in the atmosphere, why they are misrepresented especially in coastal areas in the Mediterranean, the implications, and the importance of the detailed observations you present for their better prediction.

**In the introduction, the most technical paragraph is located between lines 61-68. This paragraph has been revised, and the introduction has been amended in accordance with the reviewer's suggestions as follows:**

"The Mediterranean summer is often characterized by intense water vapour fluxes driven by significant evaporation. These fluxes feed zonal and meridional air mass transport, serving as a critical source for extreme precipitation events (Ciric et al., 2018). Severe precipitation in the Mediterranean Basin relies on both remote and local sources of anomalously high surface evaporation (Winschall et al., 2014). Water vapour fluxes primarily originate from the Atlantic, North Africa, and regional seas, contributing to the elevated relative humidity observed in the troposphere, particularly during the summer months. These fluxes play a vital role in the Mediterranean's atmospheric and hydrological systems, shaping precipitation patterns, extreme weather events, and overall climate dynamics (Drumond et al., 2018). Such fluxes have profound effects on the regional climate, intensifying rainfall in some areas while prolonging droughts in others. Persistent water vapour fluxes also influence the region's radiative balance, exacerbating surface radiation trapping and amplifying the effects of heatwaves. Consequently, understanding the role of water vapour fluxes in the Mediterranean is essential for predicting extreme weather events, such as the severe floods and heatwaves that frequently impact the region (Russo et al., 2017).

As a recognized climate change hotspot, the Mediterranean is particularly sensitive to changes in atmospheric moisture, which significantly impact weather systems, including the intensification of extratropical cyclones. However, water vapor fluxes are often poorly represented in weather and climate models, especially near coastal regions, due to the complexities involved in resolving mesoscale processes, convection and boundary layer dynamics, land-sea transitions, and aerosol interactions with moisture transport (Voulgarakis et al., 2018). These modeling challenges introduce biases in predicting extreme weather events.

Recent advances in observational programs have underscored the importance of accurately representing water vapor fluxes in forecasting severe weather. For example, marine flow-driven water vapor transport has been identified as a key factor in Mediterranean high precipitation events (HPEs), highlighting the need for realistic spatiotemporal variability of these fluxes in numerical weather prediction models (Lee et al., 2018). Furthermore, studies have shown that convection initiation in cloud-resolving models can be accurately predicted when water vapor estimates within and above the boundary layer are sufficiently detailed (e.g., Ducrocq et al., 2002; Bielli et al., 2012).

During the summer of 2021, one of the warmest on record for Europe in recent decades (Lhotka and Kyselý, 2022), several Mediterranean regions experienced severe soil moisture deficits. Southwestern Europe faced heatwaves in June, July–August, and September, with monthly average temperatures slightly below those of the warmest summer in Europe in 2022, as reported in the European State of the Climate (ESOTC) and recent studies. However, the warmest temperatures on record were observed in South Italy (ESOTC, 2023; Gandolfi et al., 2024; Merlone et al., 2024). Dry conditions in the northern Mediterranean basin extended into northern Tunisia, while soil moisture levels in other parts of northern Africa along the coastline were average to above average compared to climatological values (ESOTC, 2021).

This study investigates the effects of enhanced water vapor fluxes driven by intense Mediterranean Sea evaporation at a coastal site during the summer of 2021, using a combination of ground-based measurements from the mobile facility of the Atmospheric Observatory of the Institute of Methodologies for Environmental Analysis of the Italian National Research Council (CNR-IMAA), CIAO (Madonna et al., 2010), and the fifth-generation reanalysis data, ERA5 (Hersbach et al., 2020; Essa et al., 2022). The measurements were collected as part of the Mediterranean Experiment for Sea Salt and Dust Ice Nuclei (MESSA-DIN) in Soverato, South Italy (Latitude: 38.6894°N, Longitude: 16.545278°E, 30 m a.s.l.).

One of the primary objectives of the campaign was to study aerosol-water vapor-cloud interactions, with particular emphasis on sea salt and dust. A ground-based remote sensing facility operated at the coastal site from June 24 to November 8, 2021. This paper focuses on water vapor measurements collected until September 30, using a microwave radiometer and an infrared thermometer, which frequently recorded high relative humidity values in the mid-troposphere. Through a synergistic data analysis that incorporated the radiometer, sun photometer, aerosol lidars, and cloud radar, this study examined water vapor transport in the troposphere and compared the observations with ERA5 reanalysis data. The analysis also explored the role of aerosols in the scarcity of warm and cold cloud layers observed during the campaign. Finally, this paper highlights the potential correlation between enhanced tropospheric water vapor content from anomalous fluxes in the Mediterranean basin and severe weather events, such as the flooding that occurred in Central and Eastern Europe in July 2021. These findings emphasize the critical importance of increasing ground-based water vapor measurements across the Mediterranean region to support accurate flux predictions and improve the forecasting of extreme rainfall and flooding events.

Section 2 of the paper provides an overview of the instruments and datasets employed during MESSA-DIN. Section 3 presents the analysis of the ground-based measurements and related results, including comparisons with reanalysis data, a synoptic study of water vapour transport using ERA5, and an investigation of the role of aerosols in local cloud formation. The final section discusses the results and presents the conclusions of the study."

Added references:
- Bielli, S., Grzeschik, M., Richard, E., Flamant, C., Champollion, C., Kiemle, C., Dorninger, M., and Brousseau, P.: Assimilation of water-vapour airborne lidar observations: impact study on the COPS precipitation forecasts, Q. J. Roy. Meteor. Soc., 138, 1652–1667, 2012.
- Ducrocq, V., Ricard, D., Lafore, J. P., and Orain, F.: Storm-scale numerical rainfall prediction for five precipitating events over France: On the importance of the initial humidity field, Weather and Forecast., 17, 1236–1256, 2002.
- Lee, K.-O., Flamant, C., Duffourg, F., Ducrocq, V., and Chaboureau, J.-P.: Impact of upstream moisture structure on a back-building convective precipitation system in south-eastern France during HyMeX IOP13, Atmos. Chem. Phys., 18, 16845–16862, https://doi.org/10.5194/acp-18-16845-2018, 2018.
- Voulgarakis, A. et al., Atmospheric Chemistry and Physics, 2018. On aerosol and greenhouse gas forcing impacts on Mediterranean precipitation dynamics.
- Drumond, A. et al., Water, 2018. Contributions of the Mediterranean to continental precipitation during extreme events.

3) The title of the MS could be more specific, possibly naming the location or the campaign itself. Even the scope of the results discussed throughout is larger than suggested by the title. Furthermore, discussion of the water-vapor fluxes is brief and does not provide new insights as for the well- known importance of these fluxes for extreme weather events. The title suggests that the water vapor fluxes were derived from observation, but that's not the case. Therefore, I suggest rephrasing the title to describe the MS more accurately.

**To better align the title with the content of the manuscript, the new title will be: 'Tropospheric water vapour enhancement from Mediterranean sea fluxes during summer 2021 in Soverato (Italy): observations, comparison with ERA5, and implications for extreme events.' Additionally, the discussion of water vapour fluxes in the revised manuscript will be expanded to emphasize their importance in the context of extreme weather events.**

4) Some of the figures (4, 6, and 14) show raw data, with highly technical captions that are not intuitive for non-expert readers. Seeing that the focus of ESD is usually not set to such levels of technicality, I recommend adding a line to the captions that recalls the physical meaning of the observed quantity.

**We will add an explanatory line to the captions of Figures 4, 6, and 14 to highlight the physical meaning of the quantities represented. Additionally, we will reduce the technical level of the text, simplifying the description of the results where possible, without compromising scientific rigor.**

5) The summary should highlight the importance of the field campaign: what new information was gained and how can it be harnessed to improve our understanding or the model performance? Simply pointing out the biases seems like an underreaching conclusion, especially when comparing measurements to global reanalysis data that does not even have a corresponding grid point in the location of interest.

**At lines 423-433, we already stressed the importance of profiling measurements for validating and improving forecasts and modelling activities, putting emphasis also on the most extreme events. However, these lines, and the following, have been amended according to the reviewer's recommendations as follows:"Despite the coarse resolution of MWPs in the free troposphere, the analysis presented confirms how atmospheric profilers can be extremely valuable for validating and improving forecasting and modeling activities, especially under extreme or anomalous meteorological conditions. MWPs, for example, provide continuous measurements of temperature and water vapor in the atmosphere. In conjunction with upper-air soundings and Raman lidars, which offer higher vertical resolution, they can significantly enhance our understanding of water vapor fluxes, particularly in regions like the Mediterranean Basin.**

**Given the high spatial variability of water vapor, it is essential to conduct measurement campaigns with multiple ground-based stations to improve model validation and effectively address this variability. Despite the computational challenges, high-resolution forecasts at the kilometer scale, which can better capture mesoscale phenomena and resolve key processes like convection (rather than parameterizing them), should be beneficial for predicting extreme events. This has been demonstrated in recent studies (Caldas-Alvarez et al., 2022; Fosser et al., 2024; Chang et al., 2024) and is already being operationalized in early warning systems and meteorological services, such as the Limited Area Ensemble Prediction System (COSMO-LEPS) developed by the COSMO consortium and the German National Meteorological Service (DWD).**

**The results of the measurement campaign further highlighted the need for accurate water vapor measurements from sufficiently dense networks, particularly in regions like the Mediterranean, where the complexity of water vapor fluxes arises from the merging of remote and local sources. Current atmospheric models still struggle to reproduce the variability of water vapor due to the complexity of the underlying processes. However, these models could significantly benefit from the assimilation of high-quality measurements, which would reduce biases relative to observations and enhance their ability to predict the intensity of extreme events.**

**While reanalysis successfully replicated observations of water vapor layering and temporal evolution in the troposphere during the MESSA-DIN campaign, they still face challenges in estimating the presence of both warm and cold clouds. Aerosols played a crucial role in this context, particularly the coarse aerosol fraction associated with the transport of marine aerosols and, to a lesser extent, Saharan aerosols. These aerosols may have inhibited cloud formation, particularly in conjunction with the stable atmospheric conditions that dominated the campaign period. This further underscores the importance of improving model and reanalyses, particularly in terms of cloud formation parameterization and the representation of aerosols, especially for their type and size distribution. This study also underscores the need to improve model parameterization of the complex atmospheric processes characterizing coastal regions.**

**By improving the accuracy of water vapour flux predictions, we can better anticipate and mitigate the impacts of extreme weather events and climate variability in the Mediterranean. Persistent water vapour fluxes also impact the radiative balance, exacerbating the effect of surface radiation trapping, and potentially amplifying the effects of the heatwaves on human beings. Instead, large amounts of water vapour becoming available under these transport events can have profound impacts, particularly, affecting Central and Eastern Europe. An example is the meteorological situation in Europe from 12 to 15 July 2021, characterized by a cutoff low-pressure system over Central Europe, supplying warm and very humid air from the Mediterranean in its rotating movement. This low-pressure system led to heavy rainfall (more than 175 mm in 48 hours regionally), resulting in extensive flooding in Western Europe (Tradowsky et al., 2023). An incorrect prediction of moisture availability in the Mediterranean may become a critical factor in forecasting such extreme rain events."**

**Besides, in the comment regarding the mismatch between the reanalysis grid and the measurement station, it is important to note that this is related to the quantification of representativeness errors. This issue will be reported in the manuscript. However, as noted in our reply to reviewer #1, the nearest ERA5 grid point is located approximately 7.8 km from Soverato (38.6894°N, 16.5453°E), with the ERA5 point at (38.75°N, 16.5°E), both over land. This proximity should help minimize errors due to the spatial mismatch and, therefore, reduce the bias with respect to the true atmospheric conditions at Soverato during the 3-months study period.**

Minor comments:
L65: "investigated to identify the contribution by the contribution of water vapour fluxes and convection" – unclear
**Please see above the new version of the introduction**
Fig.1: Consider adding a larger view pointing out the location of Soverato for geographical orientation.
**Fig.1 has been replaced by the following:**

[Figure]

**Figure 1: left panel, the Landsat image of the Mediterranean Basin with the position of Sovertato site; right panel, an aerial photo of the measurement site in the town of Soverato where the MESSA-DIN campaign was carried out. Image obtained from Google Earth (version 7.3.3.7786), Accessed on December 14, 2024, Coordinates: 38.6894°N, 16.545278°E. © Google, images © 2024 Maxar Technologies.**

L88: instruments
**OK**
L127: and smaller above.
**OK**
L148-149: this sentence seems interrupted
**The sentence has be rephrased as follows: "Additionally, a sun photometer (SP) was used to estimate: column aerosol optical depth (AOD) at different wavelengths, column particle size distribution and precipitable water vapor through the data processing made by the AErosol RObotic NETwork program (AERONET).".**
L173: ad→and
**OK**
L174: biased at
**OK**
L208: larger values
**OK**
L209: rephrase. The sentence is unclear.
**OK**
L219: highlighted that measurements
**OK**
L250-265: This fits better in the introduction

L304: noting→ to note
**OK**
L324: insisting→ persisting

**OK**

L325: of a heat wave

**OK**

L326: transporter→ transport?

**OK**

L334: contributing or that likely contribute.

**OK**

L343: representation of

**OK**

L376-380: long sentence, consider splitting.

**The sentence has been rephrased as follows: "The dominance of the aerosol coarse fraction, along with evidence for the predominant contribution of marine aerosols and desert dust, suggests that the coarse particles observed during the summer of 2021 in Soverato played a key role. Under atmospheric conditions, typically dominated by persistent high pressures over South Italy, these particles contributed to a reduced likelihood of warm cloud formation at the measurement site.".**

L428-433: long sentence with several errors. Please rephrase.

**The sentence has been rephrased as follows: "In light of the resolution and forecast models, having a km-scale forecast can significantly improve the ability to address mesoscale phenomena. Despite the computational challenges, this approach allows for the resolution of important processes such as convection, rather than relying on parameterization. Recent studies (Caldas-Alvarez et al., 2022; Fosser et al., 2024; Chang et al., 2024) have shown that this is beneficial for extreme events. Furthermore, such models have already been implemented in some early warning systems and meteorological services, including the Limited Area Ensemble Prediction System (COSMO-LEPS) developed by the COSMO consortium and the German National Meteorological Service (DWD)."**